# Recent Insights into PARP and Immuno-Checkpoint Inhibitors in Epithelial Ovarian Cancer

**DOI:** 10.3390/ijerph19148577

**Published:** 2022-07-14

**Authors:** Antonios Revythis, Anu Limbu, Christos Mikropoulos, Aruni Ghose, Elisabet Sanchez, Matin Sheriff, Stergios Boussios

**Affiliations:** 1Department of Medical Oncology, Medway NHS Foundation Trust, Windmill Road, Gillingham ME7 5NY, Kent, UK; antonios.revythis@nhs.net (A.R.); a.limbu@nhs.net (A.L.); aruni.ghose@nhs.net (A.G.); elizabet.sanchez@nhs.net (E.S.); 2St. Lukes Cancer Centre, Royal Surrey County Hospital, Egerton Rd., Guildford GU2 7XX, Surrey, UK; christos.mikropoulos@nhs.net; 3Department of Medical Oncology, Barts Cancer Centre, St. Bartholomew’s Hospital, Barts Health NHS Trust, London KT1 2EE, UK; 4Department of Medical Oncology, Mount Vernon Cancer Centre, East and North Hertfordshire NHS Trust, London KT1 2EE, UK; 5Centre for Education, Faculty of Life Sciences and Medicine, King’s College London, London SE5 9NU, UK; 6Department of Urology, Medway NHS Foundation Trust, Windmill Road, Gillingham ME7 5NY, Kent, UK; matin.sheriff@nhs.net; 7Faculty of Life Sciences & Medicine, School of Cancer & Pharmaceutical Sciences, King’s College London, London SE1 9RT, UK; 8AELIA Organization, 9th Km Thessaloniki—Thermi, 57001 Thessaloniki, Greece

**Keywords:** ovarian cancer, BRCA mutations, homologous recombination deficiency, PARP inhibitors, immune checkpoint inhibitors, vaccines, adoptive immunotherapy

## Abstract

Ovarian cancer is one of the most common gynecologic cancers and has the highest mortality rate of any other cancer of the female reproductive system. Epithelial ovarian cancer (EOC) accounts for approximately 90% of all ovarian malignancies. The standard therapeutic strategy includes cytoreductive surgery accompanied by pre- or postoperative platinum-based chemotherapy. Nevertheless, up to 80% of the patients relapse within the following 12–18 months from the completion of the treatment and then receive first-line chemotherapy depending on platinum sensitivity. Mutations in *BRCA1/2* genes are the most significant molecular aberrations in EOC and serve as prognostic and predictive biomarkers. Poly ADP-ribose polymerase (PARP) inhibitors exploit defects in the DNA repair pathway through synthetic lethality. They have also been shown to trap PARP1 and PARP2 on DNA, leading to PARP-DNA complexes. Olaparib, rucaparib, and niraparib have all obtained Food and Drug Administration (FDA) and/or the European Medicine Agency (EMA) approval for the treatment of EOC in different settings. Immune checkpoint inhibitors (ICI) have improved the survival of several cancers and are under evaluation in EOC. However, despite the success of immunotherapy in other malignancies, the use of antibodies inhibiting the immune checkpoint programmed cell death (PD-1) or its ligand (PD-L1) obtained modest results in EOC so far, with median response rates of up to 10%. As such, ICI have not yet been approved for the treatment of EOC. We herein provided a comprehensive insight into the most recent progress in synthetic lethality PARP inhibitors, along with the mechanisms of resistance. We also summarised data regarding the role of immune checkpoint inhibitors, the use of vaccination therapy, and adoptive immunotherapy in treating epithelial ovarian cancer.

## 1. Introduction

Ovarian cancer is the fifth-highest cause of cancer death in the United States of America (USA) and the highest in the female reproductive system. It is estimated that about 19,880 will be diagnosed with ovarian cancer in the USA, according to the American cancer society [1]. Nevertheless, mortality is estimated to be reduced by 13% in the European Union in 2022 [2]. This reduction is believed to be due to the increased use of the oral contraceptive pill, the earlier diagnosis, and the advancements in the management of ovarian cancer. Ninety percent of ovarian cancer are of an epithelial cell type and comprise multiple histologic types, with various specific molecular changes, clinical behaviours, and treatment outcomes [3]. Malignant epithelial ovarian cancer (EOC) includes five main histological subtypes: high-grade serous ovarian cancer (70–80%), endometrioid (10%), clear cell (10%), mucinous (3%), and low-grade serous (<5%) [4,5]. Each subtype behaves as a distinct disease with differences in clinical presentation, mutations, and response to treatment such as chemotherapy [6]. It is well established that EOC develops according to two different carcinogenic pathways. Type I EOC are suggested to be relatively indolent and genetically stable tumours that typically arise from recognisable precursor lesions, such as endometriosis or borderline tumours with low malignant potential. Type I EOC includes most endometrioid, clear cell, and mucinous carcinomas. In contrast, type II EOC is proposed to be high-grade, biologically aggressive tumours from their outset, with a propensity for metastasis from small-volume primary lesions. The vast majority of EOC are high-grade serous tumours that develop according to the type II pathway and present *p53* and *BRCA* mutations. In contrast, low-grade serous tumours are characterised by *KRAS*, *BRAF*, *PTEN*, *PIK3CA*, *CTNNB1*, *ARID1A*, and *PPP2R1A* mutations and progress according to the type I pathway [7]. The remaining 10% are non-epithelial ovarian cancers, which include mainly germ cell tumours, sex cord-stromal tumours, and some extremely rare tumours [8,9]. Finally, ovarian carcinosarcomas, accounting for only 1–4% of all ovarian cancers, are composed of an epithelial as well as a sarcomatous component [10].

Newly diagnosed high-grade serous EOC patients are treated with radical surgery, followed by adjuvant platinum and taxane combination chemotherapy. In EOC patients where upfront surgery is contraindicated for medical reasons or where complete cytoreduction cannot be achieved, neoadjuvant chemotherapy is an alternative therapeutic option [11]. Treatment of gestational ovarian malignancies depends on histology, stage, and gestational weeks [12].

Genomic DNA is continuously confronted with a large number of DNA lesions. In order to keep the genome stable and secure cellular homeostasis, it is essential for the cells to counteract DNA damage by activating the DNA damage response (DDR) [13]. Several DNA damage repair pathways have evolved in cells to repair different types of damage [14]. A fourth to a fifth of unselected EOC patients carry pathogenic variants in a number of genes, the majority of which encode proteins involved in DNA damage repair pathways [3]. *BRCA1* and *BRCA2* mutations are amongst the most significant molecular alterations in EOC, breast, and prostate cancer [15,16]. Mutations in *BRCA1* and *BRCA2* cause a 20–40% risk for EOC [3]. Apart from *BRCA1* and *BRCA2*, other genes which encode for proteins involved in the homologous recombination (HR) pathway include the Fanconi anemia genes (*PALB2*, *PRIB1*), the core *RAD* genes (*RAD51C*, *RAD51D*), and genes involved in the HR pathway either directly (*CHEK2*, *BARD1*, *NBN*, *ATM*) or indirectly (such as *CDK12*, which encodes a cyclin-dependent kinase involved in the transcription regulation of several HR genes) [3]. There is an urgent need to better understand how the genomic and epigenomic heterogeneity intrinsic to EOC is reflected at the protein level and how this information could potentially lead to prolonged survival [17].

Genomic alterations in the DNA damage repair pathway are emerging as novel targets for treatment across different cancer types, especially EOC, breast, and prostate cancer. Platinum compounds and poly (ADP-ribose) polymerase (PARP) inhibitors are currently the two main classes of drugs active against cancer cells harboring DDR alterations. There are five PARP inhibitors available in clinical practice—olaparib, rucaparib, niraparib, talazoparib, and veliparib. Among them, olaparib, rucaparib, and niraparib have been approved by the Food and Drug Administration (FDA) and/or the European Medicine Agency (EMA) in EOC in different settings [18]. In 2018, talazoparib—a highly potent PARP-1 and PARP-2 inhibitor—obtained FDA approval for patients with epidermal growth factor receptor 2 (HER2)-negative metastatic breast cancer with *BRCA* mutations, relapsing after previous chemotherapy. Olaparib, rucaparib, and niraparib trap PARP approximately 100-fold more efficiently than veliparib. However, the most potent PARP inhibitor is talazoparib [19]. This is based on its enhanced capability to trap the PARP-DNA complexes at sites of single-strand DNA break damage (SSB). The difference in PARP trapping is a predictor of cytotoxicity in BRCA-mutant cells. They are also correlated with increased toxicity when combined with chemotherapeutic agents. In this regard, veliparib has a lower PARP trapping capability and can be combined with chemotherapy and radiotherapy [20].

The efficacy of immunotherapy was demonstrated in many tumours. Immune checkpoint inhibitors (ICI) are novel agents that exert immunostimulatory effects by targeting cytotoxic T-lymphocyte-associated protein 4 (CTLA-4) or programmed death-ligand 1 (PD-L1)/programmed cell death protein 1 (PD-1) [21]. PD-L1 binds to its receptor PD-1, expressed by CD4+, CD8+ T lymphocytes, and dendritic cells. PD-L1 is expressed in various cell types, including activated lymphocytes, fibroblasts, tumour-associated macrophages, and tumour cells. Based on the fact that the presence of CD3+ and intraepithelial CD8+ tumour-infiltrating lymphocytes (TIL) correlates with the survival and progression of ovarian cancer, it is considered that there is a role for modulation of the immune system in EOC [22]. Indeed, there is evidence that CD3+ TIL are associated with longer survival in advanced EOC, supporting the rationale that ICI may be a promising therapeutic approach for EOC [23]. However, early clinical trials demonstrated a response rate of only 10–15%, along with the development of drug resistance. As such, ICI are not currently approved by either FDA or EMA. As PD-L1 expression remains rare in EOC, it is necessary to further investigate potential predictive biomarkers for ICI and elucidate the key mechanisms regulating immune suppression in EOC [24].

This review article aims to highlight the recent advances in the management of EOC arising from the novel targeted molecular therapies, focusing on PARP and ICI.

## 2. DNA Damage Repair Pathways

DNA damage is a frequent event during cell life and can be spontaneous or caused by cell metabolism or by environmental agents. Six primary pathways of DNA repair have been identified, which are variably used to address double-stranded DNA break damage (DSB) and SSB from a variety of mechanisms of injury [25]. HR and nonhomologous end joining (NHEJ) are the two major pathways responsible for repairing DNA DSB, whereas the primary mechanisms for resolving DNA SSB are base excision repair (BER), nucleotide excision repair (NER), mismatch repair (MMR), and the translesional synthesis [26].

HR occurs in the S and G2 phases of the cell cycle, as it requires a template of a sister chromatid and repairs the DNA damage error-free. NHEJ is faster than HR and occurs throughout the cell cycle, but especially in the G1 phase; nevertheless, it is prone to error [26]. Meiotic recombination 11-Like (MRE11) is a nuclease that leads to the degradation of replication fork protection in the absence of BRCA proteins. In HR repair, MRE11 forms a complex with RAD50 and NBS1 (Nijmegen breakage syndrome 1), which leads to ATM recruitment and activation of the RAD3-related ATR kinase [27]. This halts the cell cycle to process DNA repair and create 3′-single-stranded DNA (ssDNA) ends [28]. The exposed ssDNA is coated by Replication Protein A (RPA), which is replaced by RAD51 for the completion of the recombinase reaction [29]. This leads to the formation of nucleoprotein filaments in ssDNA, which are essential for the homology search in sister chromatid and strand exchange [30].

In NHEJ, both free ends of DSBs are recognised and bound by the Ku70/80 heterodimer. This recruits the DNA-PK catalytic subunit to create a multi-unit complex, which then recruits further proteins, such as Artemis and PNK, to repair into a normal DNA structure [31]. Polymerases bind the broken ends, and ligases seal both DNA strands [32].

From the therapeutic point of view, targeting the DDR pathway is a reasonable approach. Within this context, several PARP inhibitors were considered for the treatment of several malignancies, including EOC, breast, and prostate cancer. The synthetic lethality of PARP inhibitors is directed against *BRCA* mutations. Technically, they are able to prevent the repair of SSB and facilitate the formation of DSB. Since *BRCA* mutant cells are deficient in the HR repair mechanism, the simultaneous inhibition of DNA repair induced by PARP inhibitors can cause cell death through apoptosis. PARP inhibitors not only block the enzymatic activity of PARP but they trap PARP1 on the damaged DNA, resulting in stalled replication forks and subsequent formation of DSB [3,18,20]. In vitro data suggested that the clinical efficacy of PARP inhibitors is associated mostly with their PARP-trapping efficiency and that mutations in PARP1 that affect its trapping can result in drug resistance [19].

## 3. PARP Inhibitors

PARP inhibitors have transformed the treatment landscape of patients with EOC. PARPs are a family of 17 nucleoproteins. Among them, PARP-1 is the best characterised and accounts for approximately 90% of the total PARP activity. PARPs are characterised by a common catalytic site that transfers an ADP-ribose group on a specific acceptor protein using NAD+ as a cofactor. Polymerisation of ADP-ribose (PARylation) is crucial for the important functions of PARP enzymes in the DNA damage response and nucleosome remodeling (Figure 1) [33]. Activation of PARPs occurs through DNA binding via zinc fingers. PARP-1, -2, and -3 are integral in the DNA damage response system by activating response pathways and facilitating repair. Unrepaired SSB or a damaged base can block the replication forks, resulting in fork collapse and DSB.

Poly ADP-ribose polymerase (PARP) binds the damaged DNA, becomes active, and catalyses the formation of PAR polymers on a variety of protein acceptors, including itself. Electrostatic repulsion between the newly formed polymer and DNA causes the release of PARP, thereby inactivating it. The poly (ADP-ribose) glycohydrolase (PARG) enzyme degrades the PAR, thereby allowing for PARP to once again bind to damaged DNA and initiate “PARylation”.

PARP inhibitors have changed the therapeutic strategy of patients with BRCA-related EOC. These agents have many similarities, but at the same time, notable differences, which are based on the differences between their chemical structural features [18]. All of the PARP inhibitors that were developed in EOC are PARP-1 and PARP-2 inhibitors, while olaparib and rucaparib additionally inhibit PARP-3. Furthermore, rucaparib inhibits tankyrase-1, which is another member of the PARP family. Currently, novel agents are in clinical development. Alterations in *BRCA* genes may also be the result of either somatic mutations or epigenetic silencing in sporadic EOC, which extends the activity of PARP inhibitors to a greater subset of sporadic EOC patients with HR deficiency. It has not yet been clarified whether the biological effects of harbouring somatic *BRCA* mutations, a phenomenon termed as BRCAness, is identical to their germline counterparts. However, there are reports of patients with somatic *BRCA* mutations who achieved longer progression-free survival (PFS) than wild-type cohorts, similar to the population with germline *BRCA* mutations. Nevertheless, OS was not affected significantly [34].

It is well recognised that when mutations occur within DNA repair pathways, there is an increased risk of malignant transformation and chemotherapy resistance. Much research has focused on protecting cells from DNA damage and/or restoring DNA repair function. However, emerging data suggest that the concept of synthetic lethality may be a particularly attractive therapeutic approach. Within this concept, the potential of PARP inhibitors therapy in EOC was highlighted by preclinical studies and clinical trials, demonstrating their superior efficacy over traditional chemotherapies. PARP inhibitors have greater specificity and fewer off-target side effects than chemotherapy or radiotherapy and can lead to more favorable clinical outcomes. They were found to have a minimal toxic impact on normal cells with functional HR.

A serious concern for the use of PARP inhibitors is the development of acquired drug resistance and *de novo* malignancies. A better understanding of how different PARP inhibitors are activated to perform overlapping and non-overlapping functions is warranted. Equally, it is important to know how PAR and NAD+ levels are modulated to alter specific biological events toward cellular survival or death. That would potentially result in the design of novel PARP inhibitors that are more specific and tumour-selective and also in the development of better strategies using reliable predictive biomarkers for the treatment with PARP inhibitors. Finally, there is a huge need to identify on which occasions we can re-sensitise recalcitrant tumour cells to PARP activity inhibition.

### 3.1. Olaparib

Olaparib is an inhibitor of PARP-1, PARP-2, and PARP-3 that was first licenced by the FDA for the treatment of patients with advanced EOC and known or suspected *BRCA* mutations, such as the fourth line of treatment back in 2014 [35]. At the same time, olaparib was licenced by EMA as monotherapy in relapsed *BRCA* mutated high-grade EOC with previous complete or partial response to platinum-based chemotherapy. This was supported by a phase II clinical trial, which compared olaparib to a placebo [36]. This study showed statistically significant higher PFS of 11.2 months in olaparib maintenance therapy, over 4.3 in placebo (hazard ratio [HR] 0.18; 95% confidence interval [CI] 0.10–0.31; *p* < 0.0001). In the same study, adverse events were described in relation to olaparib. These included nausea, fatigue, anemia, and neutropenia in a grade of equal or more than 3 in 2%, 7%, 5%, and 4%, respectively, over 0%, 3%, 1%, and 1% in the placebo group. Vomiting, taste alteration, and anorexia were also described. Multiple studies were performed on the use of olaparib in various scenarios. The SOLO1 study (NCT01844986) attempted to assess the role of maintenance olaparib in patients with ovarian cancer and germline BRCA mutations after frontline chemotherapy. The subsequent SOLO2 study (NCT01874353) investigated the use of maintenance olaparib in a similar population after two or more lines of chemotherapy. The primary endpoint was PFS, which was higher in the olaparib group with 19.1 months over 5.5 months in the placebo group (HR 0.30; 95% CI 0.22–0.41; *p* < 0.0001). The results of the SOLO2 study confirmed the findings with an impressive 30.2 months over 5.5 months of progression free survival in the olaparib group over the placebo group, respectively (HR 0.25; 95% CI 0.18–0.35, *p* < 0.0001) [37]. In the SOLO 3 study (NCT02282020), olaparib was compared as monotherapy to single-agent chemotherapy. This was a phase III randomised study in patients with recurrent EOC, with germline *BRCA* mutations that failed in two or more lines of treatment. Additionally, pooled data of 273 patients in two phase I and four phase II studies, where olaparib was used in women with relapsed disease, showed a 36% objective response rate and a 7.4-month median duration when olaparib was given after three or more lines of chemotherapy [38]. The use of PARP inhibitors is not limited to the *BRCA* mutated EOC. In 2018, olaparib and talazoparib obtained approval for HER2 negative, *BRCA1* or *BRCA2* mutated, locally advanced, or metastatic breast cancer. As far as the EOC is concerned, approximately one-third of the patients develop ascites throughout the course of the disease. Even though the treatment of the underlying disease is expected to resolve the ascites, the development of chemoresistant disease results in intractable ascites. The increased fluid production from both the tumour cells and tumour-free peritoneum, combined with compromised draining due to obstructed lymphatics, results in ascites’ buildup. Vascular endothelial growth factor (VEGF) has been shown to play a role in the formation of malignant ascites by increasing vascular permeability. Therefore, inhibition of VEGF can prevent ascites accumulation. At the same time, PARP1 plays a role in angiogenesis and can decrease VEGF expression. Inhibition of PARP1 and PARP1 knockouts has shown a decrease in induction of the transcription factor HIF-1α, which upregulates VEGF expression. It would be interesting to formulate olaparib and/or talazoparib in a nanoparticle delivery system, which would allow the drug to be administered intraperitoneally. Therefore, it might be estimated in the future whether the intraperitoneal delivery of olaparib and/or talazoparib could potentially decrease VEGF expression in the peritoneum and subsequently decrease the production of fluid.

### 3.2. Niraparib

Niraparib is an orally administrated, selective PARP-1 and PARP-2 inhibitor. Its main activity is via synthetic lethality in tumours with loss of *PTEN* and *BRCA1* or *BRCA2* function [39]. Niraparib was used in clinical studies in patients with recurrent platinum-sensitive EOC, irrespectively of the presence of *BRCA* mutations or HR deficiencies. These showed improved PFS, especially though in patients with *BRCA* mutations. Based on the double-blind, placebo-controlled, international, phase III ENGOT-OV16/NOVA study (NCT01847274), FDA approved the use of niraparib in maintenance treatment in patients with recurrence of EOC, as long as there was a complete or partial response to platinum-based chemotherapy [40]. The study enrolled 553 patients. Among them, 203 were in the germline *BRCA* cohort, and 350 patients were in the non-germline *BRCA* cohort. Patients in the niraparib group had a significantly longer median PFS than those in the placebo group, including 21.0 versus 5.5 months in the gBRCA cohort (HR 0.27; 95% CI 0.17–0.41), as compared with 12.9 months versus 3.8 months in the non-gBRCA cohort for patients who had tumours with HR deficiency (HR 0.38; 95% CI, 0.24–0.59) and 9.3 months versus 3.9 months in the overall non-gBRCA cohort (HR 0.45; 95% CI 0.34–0.61; *p* < 0.001 for all three comparisons). Severe adverse reactions (grade 3/4) included thrombocytopenia, anaemia, neutropenia, and hypertension in 29%, 25%, 20%, and 9%, respectively. When dose modification was used, most of the hematologic adverse effects were managed [40]. The future looks promising for niraparib, as the PRIMA (NCT02655016), a phase III trial investigating niraparib as a first-line treatment in ovarian cancer, and QUADRA (NCT02354586), a phase II trial investigating the use of niraparib in patients with EOC and multiple lines of treatment are underway [41,42].

### 3.3. Rucaparib

Rucaparib is a PARP-1, PARP-2 and PARP-3 inhibitor. It has been licenced since 2016 to be used as monotherapy in patients with advanced EOC with either germline or somatic *BRCA* mutations that have received two or more lines of chemotherapy [43,44]. The activity of rucaparib in the variable genetic environment was investigated in the ARIEL 2 study (NCT01891344) [45]. The population studied were women with high-grade serous or endometrioid EOC, who had received one or more lines of platinum-based chemotherapy, were platinum-sensitive, but suffered a recurrence. Three main groups were identified; one group included patients with *BRCA* mutations, another had *BRCA* wild-type and high level of loss of heterozygosity, and lastly, *BRCA* wild-type genes with a low level of loss of heterozygosity. The median PFS in these groups following treatment was 12.8 (9.0–14.7), 5.7 (5.3–7.6), and 5.2 months (3.6–5.5), respectively. The investigators concluded that patients with *BRCA*-mutated genes had a significant benefit and that assessment of loss of heterozygosity and *BRCA* status can be used to predict which patients would benefit most from rucaparib. Another important finding of the ARIEL 2 study was the adverse reaction results. Severe (grade 3) reactions of anaemia, raised transaminases, small intestinal obstruction, and malignant neoplasm progression were identified in 22%, 12%, 5%, and 5%, respectively. The follow-up ARIEL 3 study (NCT01968213) was a double-blind, placebo-controlled, phase III trial that investigated whether rucaparib could be used as maintenance treatment in platinum-sensitive patients [46]. Again, there were three groups. The first group included patients with *BRCA*-mutated genes; the second included women with defects in the HR pathway, either via *BRCA* mutation or via high levels of heterozygosity; and the third group included patients with *BRCA*-mutated genes, *BRCA*-wild type, and either high, low or indeterminate levels of heterozygosity. The PFS in these groups were 16.6 (HR 0.23; *p* < 0.0001), 13.6 (HR 0.32; *p* < 0.0001), and 10.8 months (HR 0.37; *p* < 0.0001), respectively, whilst the placebo group scored a median PFS of 5.4 months. Severe adverse events were also studied, with notable mentions of anaemia and elevated transaminases with 18.8% and 10.5%, respectively [46]. In the ARIEL4 study (NCT02855944), the efficacy and safety of rucaparib in relapsing EOC in patients with *BRCA*-mutated genes are being investigated when compared with standard chemotherapy [47]. In the efficacy population (220 patients in the rucaparib group; 105 in the chemotherapy group), the median PFS was 7.4 months (95% CI 7.3–9.1) in the rucaparib group versus 5.7 months (5.5–7.3) in the chemotherapy group (HR 0.64; 95% CI 0.49–0.84; *p* = 0.0010). Most treatment-mediated adverse events were of grade 1 or 2.

### 3.4. Veliparib

Veliparib is another PARP-1 and PARP-2 inhibitor, which is given orally and is used either in combination chemotherapy or as monotherapy [39]. Its mechanism of action is primarily via PARP inhibition but also acts via sensitising cancer cells to DNA-damaging drugs, such as oxaliplatin, cisplatin, carboplatin, irinotecan, and cyclophosphamide, as well as radiotherapy [48]. There are various data from phase I and II studies on the use of veliparib in monotherapy for EOC with subgroups of *BRCA*-mutated genes. Notably, the phase II gynaecologic oncology group (GOG) 280 study (NCT01540565) published in 2015 demonstrated an overall response rate of 26% (95% CI 16–38%) [49]. The most common side effects were fatigue, nausea and vomiting, and leukopenia. The combination studies of veliparib with other chemotherapy agents have not been very promising so far. The randomised phase II NCT01306032 trial in EOC patients treated with cyclophosphamide or low dose cyclophosphamide with veliparib was terminated early, as neither overall response rate (11.8% versus 19.4%, respectively) nor median PFS (2.1 versus 2.3 months, respectively; *p* = 0.68) were improved with the combination [50]. Currently, a phase III GOG 3005 trial (NCT02470585) is investigating the use of veliparib with carboplatin and paclitaxel in high-grade serous EOC or patients with primary peritoneal cancer in a first-line setting [51]. The use of veliparib is not limited to EOC, though. Moreover, veliparib is able to pass through the blood–brain barrier and has shown promising results when used in combination with temozolomide in the treatment of intracranial tumours [20]. This combination, in particular, has proved to be effective in a variety of histologic malignancies such as B-cell lymphoma, lung, pancreatic, EOC, breast, and prostate cancer [52].

### 3.5. Talazoparib

Talazoparib is a PARP-1 and PARP-2 inhibitor, which is selective against *BRCA1*, *BRCA2,* and PTEN mutants. The use of talazoparib in EOC is limited. However, it was the second FDA- and EMA-approved agent for HER2 negative, *BRCA*-mutated breast cancer. In comparison to veliparib, it seems to have increased radiosensitising capacity and be a more potent PARP inhibitor [53]. In a monotherapy clinical trial (NCT01286987), where effectiveness was studied in over 100 patients with *BRCA1/2* mutations and solid tumours, talazoparib was associated with fatigue, anaemia, thrombocytopenia, and nausea [54]. However, it showed an objective response rate of 50% in a subset of *BRCA1/2* mutants with high-grade serous EOC. Unfortunately, talazoparib is not currently tested in EOC, but studies regarding its use in metastatic breast cancer are underway.

## 4. PARP Inhibitor Combination Therapies

PARP inhibitors have increasing indications of being used as monotherapy. However, resistance to PARP inhibitors and attempts to improve their efficacy has driven an attempt to combine PARP inhibitors with other agents [55,56]. Olaparib, for one, is a prime example, as it is being investigated in a randomised, open-label phase II study (NCT01081951) [57]. In this study, patients with recurrent, platinum-sensitive EOC were randomised to either olaparib with paclitaxel and carboplatin, with olaparib maintenance or paclitaxel and carboplatin without any maintenance therapy. The results showed a slight improvement in PFS in the olaparib group of about 2 months—12.2 versus 9.6 months (HR 0.51; 95% CI 0.34–0.77, *p* = 0.0012). Another combination treatment that has been shown to be effective was olaparib with an anti-angiogenic multikinase inhibitor, cediranib. The median PFS in the combination group was 17.7 months compared to olaparib monotherapy (HR 0.42; *p* = 0.005) [43,58]. Expectedly, the side effects in the combination group were more common than in the olaparib monotherapy group. This was replicated in other phase I clinical trials of PARP inhibitors and chemotherapeutic agents such as temozolomide, cisplatin, carboplatin, gemcitabine, paclitaxel, or topotecan. These, however, were mainly in the form of myelosuppression [59,60]. Another two studies with promising results were published recently. These were phase I studies of combination therapies of olaparib with the P13K inhibitor BKM123 (NCT0101623349) and the AKT inhibitor AZD5363 (NCT02208375), respectively [61,62]. There are a number of combination studies underway, such as a trial of niraparib and pembrolizumab (TOPACIO study, NCT02657889) and niraparib with bevacizumab (ENGOTOV24/AVANOVA study, NCT02354131) [63,64]. Table 1 depicts the most important clinical trials of PARP inhibitors in EOC.

## 5. Mechanisms of PARP Inhibitors Resistance

Acquired resistance to PARP inhibitors is common, involving multiple mechanisms, including increased drug efflux, decreased PARP trapping, reestablishing replication fork stability (fork protection), and re-activation of HR. Among them, fork protection mechanism and restoration of HR mechanism are the two main mechanisms for PARP inhibitors resistance. In PARP inhibitors-resistant but *BRCA*-mutant ovarian cancer cells, both fork protection functions of *BRCA1/2* and HR are sequentially bypassed, and cells become increasingly dependent on ataxia telangiectasia and Rad3-related kinase (ATR) for recruitment of RAD51 onto DSB and stalled forks. As such, one mechanism of PARP inhibitor-mediated cytotoxicity is via dysregulation of replication fork reversal and/or restart. Therefore, stabilisation of replication forks may result in PARP inhibitor resistance. Fork remodelling is controlled by several chromatin remodelling proteins. The HR function is restored by secondary reversion mutations of *BRCA1*, *BRCA2,* and *RAD51* isoforms [67,68]. In patients with germline *BRCA*-mutated ovarian and breast cancer, secondary mutations that restore functional BRCA2 protein can be induced by exposure to cisplatin or PARP inhibitors [69,70].

p53-binding protein 1 (53BP1) is a chromatin-binding protein 1 that regulates DNA repair primarily by limiting long-range 5′–3′ nucleolytic digestion of DNA ends. The protection of DNA ends by the 53BP1-dependent pathway promotes physiological or pathological DSB repair by NHEJ. This is based on the fact that 53BP1 is involved in several NHEJ-driven biological processes. Among them are included the immunoglobulin class switching, the fusion of dysfunctional telomeres, and the chromosome aberrations caused by the exposure of *BRCA1*-deficient cells to PARP inhibitors.

The loss of 53BP1 reverses the cell and organismal lethality associated with mutations in *BRCA1*. Loss of 53BP1 in *BRCA1*-deficient cells restores, to some degree, HR in a manner that depends on the activation of end resection. This interaction points to a unique antagonism between *BRCA1* and 53BP1. Overall, initiating end resection is a key decision point in DSB repair pathway choice that affects the therapeutic efficacy of PARP inhibitors.

The recent detection of shieldin illustrates the difficulties involved in HR restoration during the development of PARP inhibitors’ resistance. Restored HR is newly contributed to the shieldin complex. Shieldin was recognised as a four-submits ssDNA-binding complex comprising REV7, c20orf196 (SHLD1), FAM35a (SHLD2), and FLJ26957 (SHLD3) [71]. It was shown to accumulate at the DSB site and attach to ssDNA to prevent DSB resection and accelerate NHEJ. Consequently, loss of shieldin hinders NHEJ and assists resistance to PARP inhibition in BRCA1 knock out cells due to restored HR [71,72]. This is additional evidence that shieldin acts in the same pathway as 53BP1.

Many other genes involved in HR, e.g., *BRCA1*, *BRCA2*, *RAD51,* and *MRE11,* are also involved in fork protection. Nonetheless, the process of resistance happening through the restoration of HR and fork protection is contradictory [73,74,75,76]. Deactivation of MUS81 or loss of PTIP in *BRCA*-mutant cells restores fork protection but has no impact on HR. In addition, overexpression of miR-493–5p also did not restore HR [76]. As measured by *RAD51* focus formation, restoration of HR is shown to be acquired before restoration of fork protection in a panel of isogenic olaparib-resistant BRCA1 mutant ovarian cancer cells [73]. This suggests that restoration of HR and fork protection are not connected to the mechanisms of PARP inhibitors’ resistance. Therefore, both restored HR and fork protection that should be taken into consideration when thinking of PARP inhibitors therapies as single agents as well as in combination to sustain therapeutic benefit.

## 6. Immunotherapy

PD-1 or its ligand PD-L1 is one major mechanism by which immunotherapy acts (Figure 2). PD-1 is a cell surface protein that interacts with PD-L1 expressed by tumour cells. This interaction stimulates exhaustion of peripheral effector T cells and conversion of effector T (Teff) cells to regulatory T (Treg) cells, thereby limiting immune response [77].

Pembrolizumab and nivolumab are anti-PD-1 monoclonal antibodies, while avelumab, atezolizumab, and durvalumab inhibit PD-L1. CD8+ cells and PD-L1 may not be the only relevant immune targets in ovarian cancer. Other immune subsets such as tumours-associated macrophages, cancer-associated fibroblasts, or Tregs may be crucial in mediating immune tolerance and resistance to PD-L1/PD-1 inhibition. In the IMAGYN050 trial, less than 25% of patients demonstrated >5% PD-L1 expression of immune cells [78]. In contrast, in tumours known to be immune responsive, such as non-small cell lung cancer, PD-L1 expression ranges from 24% to 60% [79]. The low response rate to PD-L1 inhibition in EOC could be in part explained by the low expression level of PD-L1 on tumour cells. This could be explained by the highly immunosuppressive tumour microenvironment of the disease and the low tumour mutational burden. There are some data to suggest that *BRCA* mutated or HR deficient EOC harbor higher levels of neoantigens, probably due to their defective DNA repair machinery. In addition, EOC demonstrates cancer-associated antigens such as NY-ESO-1, mutated p53, mesothelin, MUC-16, and SCP-1, which could drive immunogenicity. The therapeutic target of different immune subsets, such as tumours-associated macrophages, cancer-associated fibroblasts, and/or Tregs, may be relevant to making progress in cancer immunotherapy. In addition to the PD-L1/PD-1 axis, other immunosuppressive molecules—such as CTLA-4—should be considered for the development of new immunotherapies. Probably the best setting to incorporate immunotherapy is in newly diagnosed EOC, counting on a non-exhausted immune system. Finally, many studies have revealed that recurrent EOC is a “cold tumor” that always has an immunosuppressive tumour microenvironment with few tumour-associated antigens or tumour-specific antigens, resulting in insufficient recognition and eradication of cancer by the immune system [80]. Therefore, only a small subpopulation of EOC patients benefit from PD-1/PD-L1 blockade [81]. There is a great demand to improve the efficacy of PD-1/PD-L1 inhibitors in recurrent EOC patients. Following this, several strategies that would sensitise EOC to immunotherapy may include the dual immune checkpoint blockade, as well as a combination of ICI with PARP inhibitors, cytotoxic agents, radiotherapy, and/or anti-angiogenetics. CTLA-4 and PD-1/PD-L1 checkpoints act on different phases of immune activation. CTLA-4 regulates T-cell proliferation in the early phases of the immune response, whereas the PD-1/PD-L1 axis acts later in the tumour microenvironment. These differences provide the rationale for combining anti-PD-L1 with anti-CTLA4 with a supposed additive or synergic action in immune stimulation [82]. HR deficiency tumours are characterised by elevated PD-L1 expression; thus, they are likely to escape immune control. PARP inhibitors could restore a productive Th1 immune response by triggering DNA damage, especially in HR deficiency cells. In mouse models with mutations in *BRCA* genes, PARP inhibitors increased the mutational tumour load and TILs and activated the interferon-mediated pathway by synergising with ICI [82]. The anti-angiogenetic agents may enhance T cell trafficking and infiltration into the tumour microenvironment. This is the rationale behind the therapeutic strategy of the combination of ICI with antiangiogenetic agents. In preclinical models, the inhibition of VEGF signaling promoted anti-tumour immunity and enhanced the efficacy of immune checkpoint blockade. Moreover, the combination of ICI and anti-angiogenetic agents demonstrated a synergistic anti-tumour effect in vivo [82].

### 6.1. Pembrolizumab

Pembrolizumab is an anti- PD-1 antibody that is approved for the treatment of melanoma and non-small cell lung cancer by the FDA [83]. It was the first drug to gain approval in tumours with microsatellite instability. Its use in EOC is being tested in multiple studies either as monotherapy (NCT02608684, NCT02440425, NCT02537444, Keynote-100/NCT02674061) or in combination with PD-L1 antibodies (NCT02865811) or with bevacizumab and cyclophosphamide (NCT02853318) and also as a first-line chemotherapy option in combination with carboplatin and paclitaxel (NCT02520154, NCT02766582).

### 6.2. Nivolumab

Nivolumab is an anti-PD-1-antibody, which is indicated in melanoma, lung cancer, renal cell carcinoma, Hodgkin’s lymphoma, hepatocellular carcinoma, head and neck, urothelial and microsatellite instability-high colorectal carcinomas [83]. Nivolumab is currently tested in a variety of combinations in clinical trials. It is investigated as a combination immunotherapy with oregovomab, which is an anti-CA 125 antibody, in phase I/II study (NCT03100006) and with WT1 analog peptide vaccine plus montanide, which is a Freund’s incomplete adjuvant, and granulocyte–macrophage colony-stimulating factor (GM-CSF), which is a potent stimulator of dendritic cell maturation in phase I study (NCT02737787). In addition, its use in combination with bevacizumab in phase II study (NCT02873962), ipilimumab (NCT02498600, NCT02834013, NCT02923934), or with epacadostat, which is an inhibitor of indoleamine 2,3-dioxygenase, in a phase I/II study, (ECHO-204/NCT02327078) is underway.

### 6.3. Ipilimumab

Ipilimumab is a recombinant, anti-human CTLA-4 antibody that is FDA approved for the treatment of melanoma [83]. Other indications include urothelial and lung cancers. Its effect is via targeting CTLA-4 antibodies. Its use in EOC is currently being tested. In the NCT01611558 study, ipilimumab is being used in recurrent platinum-sensitive EOC as monotherapy or in combination with nivolumab.

### 6.4. Avelumab

Avelumab is a humanised monoclonal anti-PD-L1 antibody that does not block PD-1 interaction with PD-L2 [83]. It has been FDA approved since March 2017 for the treatment of Merkel cell skin carcinoma. Its use in EOC has been tested in two phase III trials. In the first one, its use as first-line therapy in combination with carboplatin and paclitaxel (Javelin ovarian 100, NCT02718417), and in the second as a treatment for recurrent platinum-resistant/refractory disease, in combination with pegylated liposomal doxorubicin versus pegylated liposomal doxorubicin alone (Javelin ovarian 200, NCT02580058). Overall response rates in PD-1 positive and negative cohorts were 11.8% and 7.9%, respectively, and the cut-off for PD-L1 positivity was set at 1% [84].

### 6.5. Atezolizumab

Atezolimumab is a humanised, monoclonal antibody that targets PD-L1 and has been approved by the FDA for the treatment of bladder/urothelial carcinomas. It has also been investigated for recurrent EOC. In a phase I study, when analysed according to biomarker status, objective response to atezolizumab was observed in 2 of 8 patients who had ≥5% PD-L1 expression, but not in the patient whose PD-L1 expression was <5% [85]. ATALANTE (NCT02891824) is a randomised, double-blinded, phase III trial evaluating the efficacy and safety of adding atezolizumab to platinum-based chemotherapy and bevacizumab [86]. The latest review in July 2017 did not identify any safety issues, and the study is currently recruiting internationally. Furthermore, there is a phase II randomised trial in patients with recurrent platinum-resistant EOC (EORTC-1508, NCT02659384), which is intended to investigate atezolizumab with bevacizumab or acetylsalicylic acid. Finally, a phase II/III randomised study (NCT02839707) is currently assessing the potency of pegylated liposomal doxorubicin with atezolizumab and/or bevacizumab. The recruitment has been completed.

### 6.6. Durvalumab

Durvalumab is a monoclonal antibody against PD-L1. Currently, it is being assessed in a phase I/II study (NCT02484404) in combination with olaparib and cediranib in advanced or recurrent EOC. Other studies include the phase I/II NCT02431559 study of durvalumab in patients with platinum-resistant recurrent EOC, scheduled to receive pegylated liposomal doxorubicin and the phase I NCT01975831 study of durvalumab and tremelimumab—a human monoclonal antibody against CTLA-4—in patients with advanced solid tumours. Both studies showed a manageable safety profile, with preliminary evidence of clinical activity. Moreover, the phase II NCT02764333 study sought to examine whether a combination of TPIV200—a multi-epitope anti-folate receptor vaccine—with durvalumab would result in enhanced anti-tumour immunity and therapeutic efficacy in patients with advanced platinum-resistant EOC. Finally, the phase I/ II NCT02726997 study is aimed to assess the pharmacodynamics and feasibility of durvalumab as a first-line treatment of EOC.

Table 2 depicts the most important ICI monotherapy trials in EOC.

## 7. Vaccination Strategies

In 1891, William Coley was the first person to use a vaccination to stimulate immune function when he injected *Streptococcus pyogenes* and *Serratia marcescens* intratumourally, after noticing regression of sarcoma in a patient with erysipelas [92]. Tumour-specific immune response could be acquired with a vaccination using various antigens [93]. Modern cancer vaccines are aimed to trigger cell-mediated immunity, which in turn actives the immune cells, and detects and abolishes the malignant cells. By using multiple techniques, the selected tumour-associated antigens are delivered against different tumours. It may be cell-based vaccines, peptide/protein, epigenetic and genetic vaccines which are either given alone or in combination with different adjuvants. e.g., cytokines or other stimulatory factors [94,95,96]. In ovarian cancer, there are multiple tumour-associated antigen molecules detected on the surface or inside the cell, which can likely serve as targets for immune recognition and response—e.g., CA125, p53 protein, FRα, HER2, and cancer-testis antigens, such as MAGE-A4 and NY-ESO-1 [92]. Currently, these vaccines are mainly pilot and phase I/II trials [100/94]. Additionally, the tumour-specific intra-nodal autologous mucin 1 targeted-dendritic cell vaccines are being tested in a phase I/II study NCT02432378 [97]. Several other trials where vaccines were tested include: (i) A phase I study (NCT01376505) on a vaccine composed of two HER2 peptides—MVF-HER-2 and MVF-HER-2, tested in various metastatic tumours, including ovarian cancer, and (ii) A study on ID-LV305 vaccine, comprising of lentiviral vector aiming dendritic cells, and consisting sequences encoding the NY-ESO-1 antigen (NCT02122861) [98]. Through DNA hypomethylation, epigenetic modulation of cancer-testis antigen genes can increase antigen expression and the potential for vaccine efficacy [99]. A randomised phase II trial that compared a dendritic cells vaccine, which targeted MUC-1 with standard-of-care in advanced EOC, reported a significantly prolonged PFS after second-line chemotherapy [97]. With additional research on the identification of possible targets, tumour vaccines may unfold as a personalized immunologic treatment for ovarian cancer.

## 8. Adoptive Immunotherapy

Cell therapies represent a personalised form of immunotherapy in which autologous lymphocytes are expanded. Adoptive T-cell therapies (ACT) can be subclassified into expanded natural TILs, T-cell receptor (TCR) and chimeric antigen receptor (CAR) T cells. ACT operate by using autologous or allogeneic anti-tumour lymphocytes to initiate cancer regression. In this technique, peripheral blood lymphocytes are separated by the process of apheresis. The precursor CD8+ lymphocytes are then stimulated with specific tumour antigens to obtain a tumour-reactive ACT product. The ACT re-infusion can be performed intravenously or intraperitoneally with recombinant interleukin-2 after lymphodepleting chemotherapy [100,101]. HLA-A2-restricted TCR specific for epitopes from known EOC antigens such as NY-ESO-1 and MAGE-A4 are currently available. NY-ESO-1 antibodies can be detected in the serum of EOC and, as such, represent an attractive target for antigen-specific ACT in EOC. Namely, NY-ESO-1 antigen-reactive TCR (retroviral vector transduced) alone or as a vaccine were tested in NCT01567891 and NCT01697527, respectively, whereas NCT02111850 investigates the anti-MAGE-A3 TCR (retroviral transduced) autologous peripheral blood lymphocytes. In a phase I study, 13 patients with EOC were treated with adoptive TIL, followed by surgery and adjuvant platinum-based chemotherapy. The 3-year survival rate was 100% in the experimental versus 67.5% in the control arm, respectively [102]. There is a concern regarding how to obtain tumour-specific lymphocytes. In order to overcome this problem, it was proposed to genetically modify T cells with TCR and CAR. Large numbers of tumour-specific T cells can be reached by introducing these TCRs targeting the NY-ESO-1 antigen [103].

T cells can also be engineered CAR to detect the tumour antigens in an MHC-independent manner as well [104]. Ongoing clinical trials are evaluating TCR or CAR-redirected T cells against NY- ESO-1 and mesothelin (NCT01583686).

## 9. Conclusions

The development of PARP inhibitors is a successful application of bench-to-bedside medicine. HR deficiency remains a strong predictor of clinical benefit from PARP inhibitors. However, *BRCA* mutations are only part of a complex group of genomic alterations leading to HR deficiency. PARP inhibitors are effective for patients with either germline or somatic *BRCA* mutations. They could also be effective in patients with HR proficient tumours. Combinations of PARP inhibitors with drugs that inhibit HR may sensitise EOC with primary or secondary HR proficiency to PARP inhibitors and expand their indication beyond HR-deficient EOC. Moreover, early-phase trials of PARP inhibitors combined with drugs targeting ATR, WEE1, and VEGF have demonstrated clinical efficacy. Finally, combinations with DNA damaging agents—namely cytotoxic agents and radiotherapy—are considered; nevertheless, the cumulative toxicities remain a serious concern. Resistance to PARP inhibitors needs to be further explored.

EOC remains one of the few malignancies in which ICI have not been incorporated into the approved standard of care. The tumour immune microenvironment is an important regulator of immune suppression and immune tolerance. It is crucial to overcome the immunosuppressive tumour microenvironment in order to improve the efficacy of ICI in EOC. Moreover, besides PD-L1, biomarkers with a predictive role in ICI should be investigated. The constitutional EOC-activated pathway may be involved in next-generation ICI. For example, the double block of TGF-beta and PD-L1 may overcome resistance to immunotherapy. Finally, the combination of PARP inhibitors and immunotherapy can improve anti-tumour immune response and enhance treatments’ efficacy.

The use of PARP inhibitors and ICI should be broadened in the upcoming years with the expectation of improving the outcome of the patients diagnosed with EOC.

## Figures and Tables

**Figure 1 ijerph-19-08577-f001:**
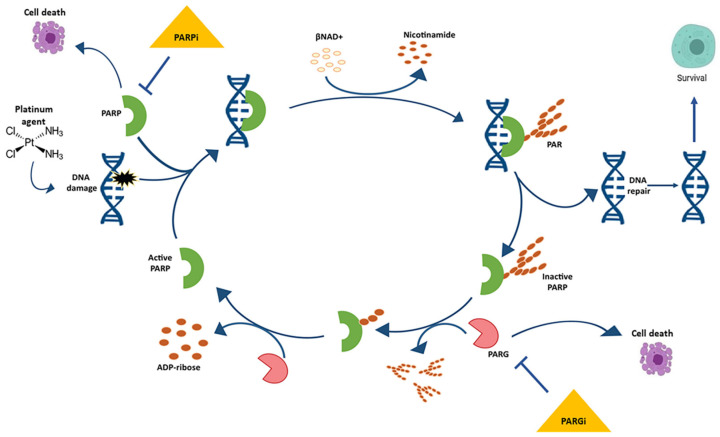
The cycle of poly ADP-ribose (PAR) metabolism “PARylation”.

**Figure 2 ijerph-19-08577-f002:**
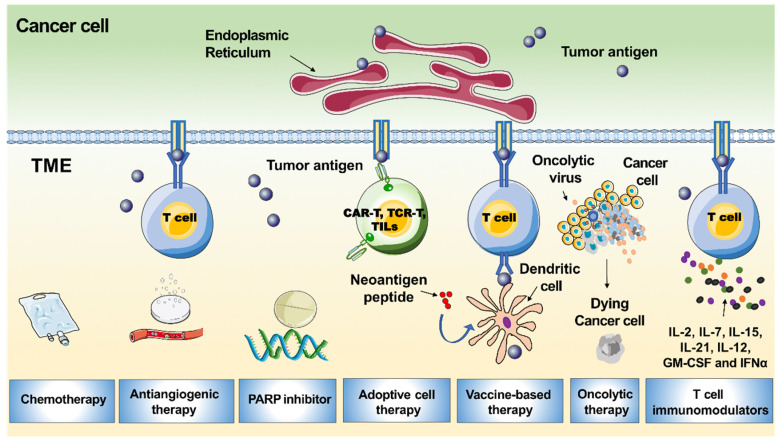
Mechanism of immunotherapy. Blocking the interaction between PD-1 and PD-L1 could reverse and/or prevent the exhaustion of tumour-specific T lymphocytes, promoting the surveillance and destruction of tumour cells. TME; tumour microenvironment.

**Table 1 ijerph-19-08577-t001:** Poly(ADP-ribose) polymerase (PARP) inhibitors trials in EOC.

Combined Treatment (Type and Pathways)	Approval	Studies/Ref	Setting	Target	PARP Inhibitor
Antiangiogenics Immunotherapy PI3K/AKT/mTOR Wee1 Chemotherapy	FDA and EMA	SOLO1, SOLO2, SOLO3/[37,65,66]	EOC with *BRCA* mutations HER2-negative BRCA-mutated breast cancer Metastatic pancreatic cancer with germline *BRCA* mutations Prostate cancer	PARP-1, -2 and -3	Olaparib
Antiangiogenics Immunotherapy	FDA	ENGOTOV16/NOVA, PRIMA, QUADRA/[40,41,42]	Platinum-sensitive EOC with *BRCA* mutations	PARP-1 and -2	Niraparib
Immunotherapy	FDA and EMA	ARIEL2, ARIEL 3, ARIEL 4/[45,46,47]	Monotherapy in advanced EOC with germline or somatic *BRCA* mutations	PARP-1, -2 and -3 and tankyrase-1	Rucaparib
Chemotherapy Radiotherapy	No	NCT01540565/[49]	EOC with *BRCA* mutations	PARP-1 and -2	Veliparib
Immunotherapy Chemotherapy	No	NCT01286987/[54]	HER2-negative breast cancer	PARP-1 and -2	Talazoparib

EOC: epithelial ovarian cancer; Ref: references; FDA: Food and Drug Administration; EMA: European Medicine Agency.

**Table 2 ijerph-19-08577-t002:** Results from trials exploring efficacy and safety of single-agent ICI in EOC.

Reference	Study/Phase	Treatment	ORR (%)	PFS (Months)	OS (Months)
JAVELIN Solid Tumour/[84]	Efficacy and safety of avelumab for patients with recurrent or refractory EOC Phase IB	Avelumab 10 mg/kg q2w	9.6	10.2	11.2
[85]	A study of atezolizumab to evaluate safety, tolerability, and pharmacokinetics in participants with locally advanced or metastatic solid tumours Phase IB	Atezolizumab 15 mg/kg q3w	22.2	2.9	11.3
KEYNOTE-028/[87]	Study of pembrolizumab in subjects with select advanced solid tumours Phase IB	Pembrolizumab 10 mg/kg q3w	11.5	1.9	13.8
KEYNOTE-100/[88]	Efficacy and safety study of pembrolizumab in participants with advanced, recurrent EOC Phase II	Pembrolizumab 200 mg q3w	8.0	1.9	13.8
[89]	Safety and anti-tumour activity of nivolumab in patients with platinum-resistant EOC Phase II	Nivolumab 1 or 3 mg/kg q2w	15	3.5	20.0
[90]	Safety and anti-tumour activity of ipilimumab in patients with EOC, previously vaccinated with GM-CSF Phase I	Ipilimumab 3 mg/kg up to 11 infusions	11.1	NR	NR
[91]	Safety and anti-tumour activity of ipilimumab in patients with recurrent platinum-sensitive EOC, previously treated with 1–4 lines of chemotherapy Phase II	Ipilimumab 10 mg/kg q3w × 4 followed by 10 mg/kg q12w	10.3	NR	NR

ICI: immune checkpoint inhibitors; EOC: epithelial ovarian cancer; ORR: objective response rate; PFS: progression-free survival; OS: overall survival; w: weeks; GM-CSF: granulocyte–macrophage colony-stimulating factor; NR: not reported.

## Data Availability

Not applicable.

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
