# Peer review of "Recent Insights into PARP and Immuno-Checkpoint Inhibitors in Epithelial Ovarian Cancer"

_ijerph, 2022, doi:10.3390/ijerph19148577_

Round 1

Reviewer 1 Report

The review 'Recent insights into PARP and immuno-checkpoint inhibitors in the ovarian cancer' provides comprehensive insight into the most recent progress in synthetic lethality drugs and immune checkpoints inhibitors in treating ovarian cancer. I suggest publishing the review with minor editing.

Major issue:

1. A paragraph from lines 151-168 focues on MMR is very confusing. The full name of MMR is provided in the first place and the content is super relevant to the rest of the review. I suggest to delete this paragraph.

2. The role of 53BP1 in restoring HR is not mentioned in the review. Since mutation 53BP1 or its downstream proteins, such as the Shieldin complex, authors mentioned in the paragraph from lines 360-366 are the key factors contributing to PARPis resistance. Therefore, I suggest the authors could review this part of the literature more carefully.

Minor issues

Please check the typo throughout the manuscript, such as in line 30, ''Immune checkpoint inhibitor (CPI)' should be 'immune checkpoint inhibitor (ICI)'.

Author Response

Dear Editor and Reviewers,

I am pleased to resubmit for publication the revised version of ijerph-1779110 manuscript, entitled “Recent insights into PARP and immuno-checkpoint inhibitors in the ovarian cancer”.

Thankfully the reviewers provided us with a great deal of guidance, regarding how to better position the article. We are hopeful you agree that this revision will update our comprehensive review. All the comments have been addressed, as shown in the revised version of the manuscript, along with this point-by-point response to the reviewers' comments.

All corresponding are blue changes in the manuscript.

Reviewer #1:

General comments:

The review 'Recent insights into PARP and immuno-checkpoint inhibitors in the ovarian cancer' provides comprehensive insight into the most recent progress in synthetic lethality drugs and immune checkpoints inhibitors in treating ovarian cancer. I suggest publishing the review with minor editing.”.

Response:

Thank you very much for your kind words about our paper. We appreciate the opportunity to revise our work for consideration for publication.

  • Major issues:

  1. A paragraph from lines 151-168 focues on MMR is very confusing. The full name of MMR is provided in the first place and the content is super relevant to the rest of the review. I suggest to delete this paragraph.

Response:

Thank you for your comment. We fully agree and we have excluded that paragraph from the revised manuscript, as you kindly recommended.

  1. The role of 53BP1 in restoring HR is not mentioned in the review. Since mutation 53BP1 or its downstream proteins, such as the Shieldin complex, authors mentioned in the paragraph from lines 360-366 are the key factors contributing to PARPis resistance. Therefore, I suggest the authors could review this part of the literature more carefully.

Response:

Thank you for your recommendation. Indeed, that part might be expanded appropriately. We have added two paragraphs to clarify the role of 53BP1 in restoring HR (Lines 409-422 on the revised manuscript).

  • Minor issue:

Please check the typo throughout the manuscript, such as in line 30, ''Immune checkpoint inhibitor (CPI)' should be 'immune checkpoint inhibitor (ICI)'.

Response:

Thank you. We have edited the typo and checked carefully throughout the manuscript.

Reviewer 2 Report

In the article with the title “Recent insights into PARP and immuno-checkpoint inhibitors in the ovarian cancer”, the authors aim to provide recent insights mainly into the use of PARP and immuno-checkpoint inhibitors in epithelial ovarian cancer. Moreover, they provide information about the use of vaccination therapy and adoptive immunotherapy. Even though the aim of this review is significant for the relative scientific field, numerous issues exist, as explained below:

Major issues

·         Title: The main part of the article includes information about epithelial ovarian cancer. Therefore, it is suggested to include the word “epithelial” in the title.

·         Abstract: The aim of the review as it is described in the Abstract is not clear. Moreover, the “…focus on the therapeutic implication of BRCA mutations.” Is not clear in the manuscript.

·         Introduction: The third paragraph could be shorter and combined with the fourth paragraph.

·         Introduction: The fifth paragraph is not properly connected with the previous.

·         Introduction: “However, the most potent PARP inhibitor is talazoparib”. It is advised to provide a further explanation of this sentence. Why it is the most potent?

·         DNA damage repair pathways: It is not clear why the authors decided to extensively refer to DNA repair mechanisms. Moreover, information about the role/ mechanism of action of PARPs is essential.

·         It is suggested to include information about the benefit of these targeted therapies in comparison to classic therapeutic strategies. Moreover, it is suggested to provide more clear information about the benefits and the disadvantages of these drugs.

·         The information about the use of a component in another type of cancer, as in the last sentence of the Olaparib section is beneficial. However, a comment with future perspectives and how this drug could be used in epithelial ovarian cancer is essential.

·         Mechanisms of PARP inhibitors resistance: What is the fork protection mechanism?

·         Immunotherapy: “The low response rate to PD-L1 inhibition in EOC could be in part explained by the low expression level of PD-L1 on tumor cells.” Then why is this strategy proposed for epithelial ovarian cancer? It is suggested that the authors properly discuss this.

·         An extended discussion with challenges and future perspectives is missing.

·         Another review article with a common topic exists: Nero, C.; Ciccarone, F.; Pietragalla, A.; Duranti, S.; Daniele, G.; Salutari, V.; Carbone, M.V.; Scambia, G.; Lorusso, D. Ovarian Cancer Treatments Strategy: Focus on PARP Inhibitors and Immune Check Point Inhibitors. Cancers 202113, 1298. https://doi.org/10.3390/cancers13061298. Therefore, it is advised to refer to this review highlighting the advantages of this new manuscript for the relative scientific field.

Minor issues

·         Abstract: According to CLOBOCAN, breast cancer has the highest mortality rate.

·         Abstract: There is a mistake in the abbreviation of immune checkpoint inhibitors.

·         Introduction: What are the two pathways which are named in the first paragraph as type I and type II. More information is needed.

·         Introduction: in line 75 the “has” needs to be corrected.

·         It is advised to provide specific information about which molecule is targeted by each drug.

·         It is highly recommended to provide a second table with information about ICIs. Additionally, Table 1 could also include the target of each drug, if it is given as a monotherapy or not, and the existence of FDA or EMA approval.

·         The legend of Figure 2 does not properly describe the Figure. It is advised to provide this Figure in the Introduction section, together with a short explanation of the main chemotherapeutic and antiangiogenic therapies which are used in epithelial ovarian cancer.

·         The last sentence of the manuscript seems extremely optimistic.

Author Response

Dear Editor and Reviewers,

I am pleased to resubmit for publication the revised version of ijerph-1779110 manuscript, entitled “Recent insights into PARP and immuno-checkpoint inhibitors in the ovarian cancer”.

Thankfully the reviewers provided us with a great deal of guidance, regarding how to better position the article. We are hopeful you agree that this revision will update our comprehensive review. All the comments have been addressed, as shown in the revised version of the manuscript, along with this point-by-point response to the reviewers' comments.

All corresponding are blue changes in the manuscript.

Reviewer #2:

General comments:

In the article with the title “Recent insights into PARP and immuno-checkpoint inhibitors in the ovarian cancer”, the authors aim to provide recent insights mainly into the use of PARP and immuno-checkpoint inhibitors in epithelial ovarian cancer. Moreover, they provide information about the use of vaccination therapy and adoptive immunotherapy. Even though the aim of this review is significant for the relative scientific field, numerous issues exist, as explained below:”.

Response:

We appreciate you taking the time to offer us your comments and insights related to the paper. Thank you for your positive reinforcement and constructive feedback. We tried to be responsive to your concerns as we approached our revision.

  • Major issues:

  1. Title: The main part of the article includes information about epithelial ovarian cancer. Therefore, it is suggested to include the word “epithelial” in the title.

Response:

Thank you for your comment, which absolutely makes sense. The title has now been modified as follows:

Recent insights into PARP and immuno-checkpoint inhibitors in epithelial ovarian cancer”.

  1. Abstract: The aim of the review as it is described in the Abstract is not clear. Moreover, the “…focus on the therapeutic implication of BRCA mutations.” Is not clear in the manuscript.

Response:

Thank you for your comment. We have now rephrased the final part of the abstract to better reflect the context of the manuscript (Lines 35-38 on the revised manuscript).

  1. Introduction: The third paragraph could be shorter and combined with the fourth paragraph.

Response:

Thank you for your recommendation. We have now modified the previous third paragraph and combined with the fourth (Lines 76-92 on the revised manuscript).

  1. Introduction: The fifth paragraph is not properly connected with the previous.

Response:

Thank you for your comment. We have now added lines 93-97 in the revised manuscript, which connected better with the previous manuscript.

  1. Introduction: “However, the most potent PARP inhibitor is talazoparib”. It is advised to provide a further explanation of this sentence. Why it is the most potent?

Response:

Thank you for your comment. We have now clarified appropriately (Lines 105-107 on the revised manuscript).

  1. DNA damage repair pathways: It is not clear why the authors decided to extensively refer to DNA repair mechanisms. Moreover, information about the role/ mechanism of action of PARPs is essential.

Response:

Thank you for your consideration. A defective DNA-damage response is a defining hallmark of high-grade serous EOC. We have decided to refer to DNA repair mechanisms as we believe that understanding the relationship between these pathways and how these are abrogated will be necessary in order to facilitate appropriate selection of both existing and novel agents.

Moreover, we have added in that section a few sentences about the synthetic lethality and the PARP-trapping (Lines 158-167 on the revised manuscript).

Finally, the first three paragraphs of the “3.PARP inhibitors” section of the original manuscript included data associated with the role/ mechanism of action of PARP inhibitors.

  1. It is suggested to include information about the benefit of these targeted therapies in comparison to classic therapeutic strategies. Moreover, it is suggested to provide more clear information about the benefits and the disadvantages of these drugs.

Response:

Thank you for your suggestion. We have incorporated the two final paragraphs in the “3. PARP inhibitors” section, comparing PARP inhibitors with chemotherapy and radiotherapy. We also provided more clear information about the benefits of PARP inhibitors and the future challenges (Lines 205-224 on the revised manuscript).

  1. The information about the use of a component in another type of cancer, as in the last sentence of the Olaparib section is beneficial. However, a comment with future perspectives and how this drug could be used in epithelial ovarian cancer is essential.

Response:

Thank you for your statement. We have made a comment with future perspectives and how olaparib and talazoparib could be used in epithelial ovarian cancer (Lines 255-271 on the revised manuscript).

  1. Mechanisms of PARP inhibitors resistance: What is the fork protection mechanism?

Response:

Thank you for your statement. We have now clarified what is the fork protection mechanism and reorganized the first paragraph of the “5. Mechanisms of PARP inhibitors resistance” section (Lines 394-408 on the revised manuscript).

  1. Immunotherapy: “The low response rate to PD-L1 inhibition in EOC could be in part explained by the low expression level of PD-L1 on tumor cells.” Then why is this strategy proposed for epithelial ovarian cancer? It is suggested that the authors properly discuss this.

Response:

Thank you for your comment. We have discussed further the reasons behind the low response rate to PD-L1 inhibition in EOC and also consider the therapeutic target of different immune subsets, such as tumors-associated macrophages, cancer-associated fibroblasts and/or Tregs. We refer to lines 464-475 and 481-483 of the revised manuscript.

  1. An extended discussion with challenges and future perspectives is missing.

Response:

Thank you for your comment. The “9. Conclusions” section of the original manuscript has now been entitled “9. Conclusions and future directions”. At that, we have incorporated lined 636-640 and 646-650 for the PARP- and the ICI, respectively.

  1. Another review article with a common topic exists: Nero, C.; Ciccarone, F.; Pietragalla, A.; Duranti, S.; Daniele, G.; Salutari, V.; Carbone, M.V.; Scambia, G.; Lorusso, D. Ovarian Cancer Treatments Strategy: Focus on PARP Inhibitors and Immune Check Point Inhibitors. Cancers 2021, 13, 1298. https://doi.org/10.3390/cancers13061298. Therefore, it is advised to refer to this review highlighting the advantages of this new manuscript for the relative scientific field.

Response:

Thank you for your statement. We have read the interesting review – as you kindly recommended – and cited it in our manuscript. We have incorporated important information associated with the immunotherapy in the section “6. Immunotherapy” (Lines 483-499 on the revised manuscript).

  • Minor issues:

  1. Abstract: According to CLOBOCAN, breast cancer has the highest mortality rate.

Response:

Thank you for your comment. We have clarified appropriately (Line 20 on the revised manuscript).

  1. Abstract: There is a mistake in the abbreviation of immune checkpoint inhibitors.

Response:

Thank you. We have edited the typo (Line 31 on the revised manuscript).

  1. Introduction: What are the two pathways which are named in the first paragraph as type I and type II. More information is needed.

Response:

Thank you. The relevant information has been added in lines 56-62 on the revised manuscript.

  1. Introduction: in line 75 the “has” needs to be corrected.

Response:

Apologies for the grammar mistake. In line ff of the revised manuscript, we have replaced “has” by “have” (Line 79 on the revised manuscript).

  1. It is advised to provide specific information about which molecule is targeted by each drug.

Response:

Thank you for you recommendation. This information had already been provided in lines 193-196. Apart from that, we have now added lines 226 and 356-357.

  1. It is highly recommended to provide a second table with information about ICIs. Additionally, Table 1 could also include the target of each drug, if it is given as a monotherapy or not, and the existence of FDA or EMA approval.

Response:

Thank you for you recommendation. Table 1 has been modified appropriately and a second table (Table 2) with information about ICIs has been added as you kindly advised.

  1. The legend of Figure 2 does not properly describe the Figure. It is advised to provide this Figure in the Introduction section, together with a short explanation of the main chemotherapeutic and antiangiogenic therapies which are used in epithelial ovarian cancer.

Response:

Thank you for you advice. Figure 2 describe the mechanism of immunotherapy. As such, we would prefer to keep it in that section (“6. Immunotherapy”) without modifications. Thank you for understanding.

  1. The last sentence of the manuscript seems extremely optimistic.

Response:

Thank you for you comment. We have now rephrased the final sentence of the manuscript (Lines 651-652 on the revised manuscript).

Round 2

Reviewer 2 Report

All previous comments have been properly addressed and the manuscript has been greatly improved. The revised version is well organized and includes all the relative information.